Vertical and horizontal distribution of Desmophyllum dianthus in Comau Fjord, Chile: a cold-water coral thriving at low pH

Fillinger Laura laura.fillinger@gmail.com
Richter Claudio
Alfred-Wegener-Institut, Helmholtz-Zentrum für Polar- und Meeresforschung , Bremerhaven , Germany
Qian Pei-Yuan
Electronic publication date: 2013 Oct 29
Publication date: 2013
Volume: 1
Electronic Location ID: e194
Received 2013 Jul 19; Accepted 2013 Oct 8
Copyright: © 2013 Fillinger and Richter
Copyright year: 2013
Copyright holder: Fillinger and Richter
License: This is an open access article distributed under the terms of the Creative Commons Attribution License, which permits unrestricted use, distribution, and reproduction in any medium, provided the original author and source are credited.
License URL: https://creativecommons.org/licenses/by/3.0/

Keywords: Ocean acidification, Desmophyllum dianthus, Cold-water corals, pH, Remotely Operated Vehicle

Funding: German Federal Ministry of Education and Research (BMBF) project “Ökologie von Hydrokorallenriffen im Flachwasser der Patagonischen Fjorde Chiles” Grant number CHL08/001 This research was supported by the Alfred Wegener Institute (PACES T1.6) with additional funding by the project “Ökologie von Hydrokorallenriffen im Flachwasser der Patagonischen Fjorde Chiles” (International Office of the German Federal Ministry of Education and Research, Grant number CHL 08/001). The funders had no role in study design, data collection and analysis, decision to publish, or preparation of the manuscript.

==============================
Cold-water corals provide an important habitat for a rich fauna along the continental margins and slopes. Although these azooxanthellate corals are considered particularly sensitive to ocean acidification, their responses to natural variations in pH and aragonite saturation are largely unknown due to the difficulty of studying their ecology in deep waters. Previous SCUBA investigations have shown an exceptionally shallow population of the cold-water coral Desmophyllum dianthus in near-surface waters of Comau Fjord, a stratified 480 m deep basin in northern Chilean Patagonia with suboxic deep waters. Here, we use a remotely operated vehicle to quantitatively investigate the distribution of D. dianthus and its physico-chemical drivers in so far uncharted naturally acidified waters. Remarkably, D. dianthus was ubiquitous throughout the fjord, but particularly abundant between 20 and 280 m depth in a pH range of 8.4 to 7.4. The persistence of individuals in aragonite-undersaturated waters suggests that present-day D. dianthus in Comau Fjord may show pre-acclimation or pre-adaptation to conditions of ocean acidification predicted to reach over 70% of the known deep-sea coral locations by the end of the century.

Introduction

Similar to their tropical counterparts and in spite of their lack of photosynthetic endosymbionts, azooxanthellate scleractinian cold-water corals (CWC) provide the structural 3-dimensional basis and habitat for a rich deep-reef associated fauna (Cairns & Stanley, 1982; Mortensen & Buhl-Mortensen, 2005; Roberts et al., 2009). With an almost circumglobal distribution (Freiwald et al., 2004; Roberts et al., 2009) these cnidarians occur mostly between 200 and 1000 m depth (Cairns, 2007), making investigations in their natural environment logistically challenging (Maier et al., 2009).

One of the main reef-building species, Lophelia pertusa Linnaeus, 1758, has been shown to be particularly sensitive to short-term temperature changes (Dodds et al., 2007), and the paucity of CWC in aragonite-limited waters (Guinotte et al., 2006) has raised great concerns regarding their fate in the context of climate change and ocean acidification (McCulloch et al., 2012; Roberts et al., 2009).

Ocean acidification, a consequence of the uptake of anthropogenic carbon dioxide (CO2) by the ocean, results in pH reduction and alteration of the seawater carbonate chemistry (see e.g., Doney et al., 2009; Feely et al., 2004). As the quantity of CO2 absorbed by the ocean increases, the saturation state of the aragonite carbonate mineral used by scleractinian corals to build their skeleton decreases, leading ultimately to its dissolution (see e.g., Caldeira & Wickett, 2003; Orr et al., 2005). Besides, as carbonate solubility increases with decreasing temperature and increasing pressure, saturation states are lower in deep and cold habitats than in shallow and warm waters (Feely et al., 2004). Models predict a progressive shoaling of the aragonite saturation horizons (ASH - the limit between undersaturation and supersaturation) (Orr et al., 2005). Hence, deep-dwelling cold-water corals are expected to be the first exposed to aragonite undersaturation (Doney et al., 2009).

To date, few experiments have been conducted on the response of CWC to ocean acidification (Guinotte et al., 2006; Roberts, Wheeler & Freiwald, 2006). Lophelia pertusa exhibited reduced calcification rates under increased CO2 concentrations in short term experiments (Form & Riebesell, 2012; Maier et al., 2009) but revealed capacity for acclimation in longer term manipulations (Form & Riebesell, 2012). In both cases, calcification rates remained positive even under aragonite undersaturation. Calcification rates in Madrepora oculata Linnaeus, 1958, another reef-building species, remained stable under reduced pH but increased under pre-industrial, higher pH levels suggesting that CWC are already suffering from ocean acidification (Maier et al., 2012).

A complementary strategy to manipulation experiments in aquaria to determine the sensitivity of corals to changes in seawater chemistry is to investigate their distribution along natural pH gradients. For zooxanthellate shallow water corals, spatial differences in coral distribution were found around volcanic CO2 seeps in Italy (Hall-Spencer et al., 2008; Riebesell, 2008) and Papua New Guinea (Fabricius et al., 2011). To our knowledge, only three corresponding studies focusing on the qualitative analysis of the presence/absence of some species were published for living CWC. In the Gulf of Mexico, Lophelia pertusa seemed restricted to aragonite saturated waters but given the lack of correlation between this species’ presence and the aragonite saturation level, Lunden, Georgian & Cordes (2013) suggested that depth or other factors were more likely to control the coral’s distribution. On an Australian seamount, two species of colonial scleractinian Solenosmilia variabilis Duncan, 1873 and Enallopsammia rostrata Pourtalès, 1878 persisted near the ASH and a third solitary species Desmophyllum dianthus Esper, 1794 was detected well below this depth in aragonite undersaturated waters (Thresher et al., 2011). Pronounced pH and aragonite saturation gradients have recently been described for Comau Fjord, Chilean Patagonia (Jantzen et al., 2013a). There, Desmophyllum dianthus is known to occur in dense aggregations in high-pH near-surface waters, together with two other cup corals: Caryophyllia huinayensis (Cairns, Häussermann & Försterra, 2005) and Tethocyathus endesa (Cairns, Häussermann & Försterra, 2005; Försterra & Häussermann, 2003). The deployment of a small Remotely Operated Vehicle (ROV) halfway down the fjord (263 m) showed the presence of D. dianthus below the ASH down to at least 255 m depth (Häussermann & Försterra, 2007; Jantzen et al., 2013a).

Desmophyllum dianthus (senior synonym of Desmophyllum cristagalli (Cairns, 1995)) is a solitary azooxanthellate scleractinian often associated with Lophelia pertusa and Madrepora oculata (Heindel et al., 2010; Reveillaud et al., 2008). It can form pseudo-colonies with younger individuals growing on the skeleton of older ones and constitutes the framework-building CWC species in coral banks off New Zealand and Chile (Cairns & Stanley, 1982; Squires, 1965). The linear growth rate of deep-sea specimens has been estimated around 0.5–2 mm yr−1 (Adkins et al., 2004) but might be as fast as 2.2–10 mm yr−1 for Chilean corals (Jantzen et al., 2013b; McCulloch et al., 2005) . This cosmopolitan species has been reported over a wide depth range from 35 to 2460 m (Cairns, 1995). In the fjords of northern Chilean Patagonia, even shallower occurrences of dense D. dianthus banks have been reported (20 m depth), with scattered individuals occasionally occurring as shallow as 8 m (Försterra & Häussermann, 2003). SCUBA studies reported D. dianthus abundances up to 1500 individuals/m2 on rocky walls and overhangs with inclination >80° (Försterra & Häussermann, 2003), but a notable absence of corals on milder slopes, perhaps due to smothering by terrigenous sediments (Försterra et al., 2005; Försterra & Häussermann, 2003). The factor limiting the upper distribution of this coral is thought to be the sharp halocline, separating the marine waters from the brackish surface layer (Cairns, Häussermann & Försterra, 2005; Försterra & Häussermann, 2003). The existence and nature of a lower limit in Comau Fjord has not yet been investigated (Häussermann & Försterra, 2007).

Individual D. dianthus have been collected in the deep-sea from waters exhibiting various pH ranging from 7.57 to 8.10 (Anagnostou et al., 2012; McCulloch et al., 2012). In Comau Fjord, a recent report (Jantzen et al., 2013a) described the presence of D. dianthus along a vertical pH gradient of 7.4–8.3 between the surface and mid-depth (230 m). Although the qualitative data suggest that this CWC is less susceptible to low pH than previously assumed (Jantzen et al., 2013a), concurrent data on D. dianthus abundance, pH and other environmental parameters that might be involved in the control of this CWC are still lacking.

To close this gap, we deployed a novel ROV in Comau Fjord to quantitatively characterize the occurrence of D. dianthus (abundance and size) and simultaneously measure several physico-chemical descriptors of its natural environment.

Our main objectives were to describe the (i) occurrence, (ii) vertical, (iii) horizontal and (iv) size distribution of D. dianthus in Comau Fjord; (v) to measure the corresponding environmental parameters along the ROV survey; and (vi) analyse the resulting distribution patterns to assess the role of pH versus other environmental factors in governing D. dianthus distribution.

Materials & Methods

Site

Comau Fjord in Northern Chilean Patagonia is a 41 km long, 4.5 km wide and about 480 m deep basin with its mouth shoaling to around 300 m depth (Fig. 1). Little is known about its hydrodynamics but the principal driving processes are probably tidal exchange and mixing in this macrotidal environment with up to 7 m tidal amplitudes (Försterra, 2009), and a thermohaline circulation sustained by high precipitation (>5000 mm/year) (Bustamante, 2009). Due to its north-south orientation, the fjord is protected from the main westerly winds, so that wind-driven advection and mixing play only minor roles (Bustamante, 2009). Freshwater entering the fjord from the rivers at its head and its eastern side (Bustamante, 2009; Soto, 2009) strongly influences the surface layer down to 10 m (Sánchez, González & Iriarte, 2011). Below the shallow brackish water layer with profound seasonal changes in temperature, the lower fully marine environment is less influenced by seasonality (Sánchez, González & Iriarte, 2011). Surrounded by steep basaltic walls covered with a temperate primary rainforest (Mayr et al., 2011), the fjord is subjected to frequent landslides and avalanches, causing a high input of terrestrial sediment to the aquatic system, which then accumulates at the bottom of the basin (Försterra, 2009).

Figure 1 Area of investigation.

(A) Overview on Patagonia showing the rugged coastline broken up by fjords and channels along southern Chile. The rectangle denotes area shown in panel B. (B) Chiloé Interior Sea linking Comau Fjord (red asterisk) to the Pacific Ocean. (C) Comau Fjord (bathymetry; courtesy of SHOA) with the location of the seven ROV dive sites. Scale bar represents 5 km. (Projection for all maps: WGS 1984 UTM Zone 18S).

ROV surveys

In February 2012 we deployed our ROV (Ocean Modules: V8 Sii) at seven sites along Comau Fjord (Fig. 1, Table 1) in order to map the horizontal and vertical distribution of Desmophyllum dianthus. The vehicle was equipped with two High Definition video cameras (oe14-502; Kongsberg), one oriented horizontally for video mapping of the benthos and the other tilted 30° downward for navigation. The video streams were captured to Compact Flash cards (.mxf, mpeg2) by a nanoFlash recorder (Convergent Design). An echo-sounder (Micron; Tritech) was mounted onto the horizontal camera, measuring its distance to the substrate. The ROV also carried a factory-calibrated Conductivity Temperature Depth profiler (CTD) (SBE19 plus; SeaBird Electronics) including pH (SBE18), oxygen (SBE43) and chlorophyll fluorescence (Eco FLRT; Wetlabs) sensors for continuous records of the physico-chemical environment during the dives. While the other sensors did not require re-calibrations, the pH sensor was kept calibrated prior to deployment with TRIS (TRIS buffer, batch 5, Nr. 58, pH 8.09 at 25°C; Oceanic Carbon Dioxide Quality Control, AG Dickson, Scripps Institution of Oceanography) (Nemzer & Dickson, 2005) and two additional buffers (WTW PL4 and PL7, DIN/NIST, pH 4.006/4.01 and pH 6.865/6.87) taking care of the corresponding temperature corrections.

Table 1 Sampling sites.

Date	Site	Code	Latitude	Longitude	Max.
depth (m)	Distance to
head (km)	
09/02/2012	Lilliguapi	LL	42° 09.661′ S	72° 36.043′ W	198	40.9	
09/02/2012	Cross Lilli	CL	42° 11.295′ S	72° 35.211′ W	115	39.5	
13/02/2012	Near Telele	NT	42° 14.935′ S	72° 30.880′ W	422	26.9	
08/02/2012	Steep Wall	SW	42° 19.889′ S	72° 27.666′ W	209	17.2	
07/02/2012	Cross Huinay	CH	42° 23.213′ S	72° 27.772′ W	282	11.9	
06/02/2012	Punta Gruesa	PG	42° 24.621′ S	72° 25.446′ W	355	8.1	
11/02/2012	Rio Bodudahue	RB	42° 27.360′ S	72° 24.826′ W	216	2.0	

The survey strategy adopted there was to carry out two vertical transects per site (straight downward and upward), adapting the vehicle’s pitch so that the horizontal camera axis was kept perpendicular to the substrate.

Physico-chemical environment

Plots of the CTD data were performed with Ocean Data View (R Schlitzer, Ocean Data View, http://odv.awi.de, 2013). In the along-fjord profiles (Fig. 2), the colour scales were adapted to highlight the structure of the marine water column, at the expense of the structure of the low-salinity surface layer above the halocline which is not relevant for the corals. The full data range was used in the colour scale of the depth profiles (Fig. 3).

Figure 2 Physico-chemical sections across Comau Fjord.

Profiles obtained during the ROV downward transects (vertical black lines) from the on-board CTD and compiled in Ocean Data View for (A) salinity, (B) temperature, (C) pH and (D) oxygen concentration. The numbers within the graphs correspond to the four water masses described in the discussion: (1) low-salinity surface layer, (2) Subantarctic oceanic water, (3) deeper water mass with longer residence time, (4) subsurface minimum in pH and oxygen.

Video analyses

Frames (.png) were extracted every 10 s from the video recorded by the horizontal camera with the software COVER (developed by the IFREMER in the frame of the EU project CoralFISH). All frames were scaled, computing their length and width from the echo-sounder measurements and the camera properties (horizontal and vertical angles of view). A coordinate system was then conceived to graphically display the position of each frame in the fjord with the CTD depth along the y-axis. The x-coordinate was attributed arbitrarily, respecting the sequence of the seven sites along the fjord and placing both transects (down and up) at each site next to each other but not overlapping. The name, scale and coordinates of all frames were combined in COVER. Then a Scilab routine (Scilab Enterprises, www.scilab.org) was run to create an individual text file called a worldfile (.pgw) for each frame describing its scale and coordinates (method similar to the one used by Jerosch et al. (2011)).

Each frame accompanied by a worldfile could then be opened and projected in a metric coordinate system in ArcGIS (ESRI). Excluding overlapping frames, we determined the surface represented by the analysed frames for each station (surface analysed), the surface of non-sediment covered rock within the analysed area (available substrate) (Fig. 6) and the amount of rock colonized by Desmophyllum dianthus (coral patch size). A patch was defined as a coral aggregation within a single frame. If those aggregations were extending over several (N ≥ 3) frames, they were described as a coral bank. Each coral was referenced and counted to determine the abundance of D. dianthus within each patch (coral abundance). When a coral was adequately positioned in the field of view, in a plane perpendicular to the camera axis, and entirely visible, its length and mid-length width were also measured. The morphology of the coral was then described using these values (coral shape = length/width).

Statistical analyses

All further calculations and statistical analyses were realized in R (www.r-project.org).

Each parameter was summed (surface analysed, available substrate) or averaged (CTD data, coral patch size, coral abundance, coral shape) for each site within 10 m depth intervals. The fraction of available substrate effectively colonized by the corals was calculated in each depth interval (% used = 100 ∗ patch size/available substrate). The data on substrate and coral distribution were imported in ArcGIS for graphical representation (Figs. 4 and 5).

To investigate eventual relationships between the coral distribution parameters (coral abundance and substrate usage) and the environmental parameters (temperature, salinity, pH, oxygen and available substrate) in Comau Fjord multiple linear regression analyses were performed.

All primary data are available at http://doi.pangaea.de/10.1594/PANGAEA.811911.

Results

Fjord environment

The low salinity surface layer was limited to the upper 7 m at all sites (Figs. 2 and 3). Below this depth, the salinity slowly increased with depth in a range of 31.2–33.0. Surface water temperatures reached up to 17°C (Table 2) but below the thermocline (20–50 m depth), the temperatures remained between 10.2 and 12.0°C, showing a slight increase with depth. Oxygen and pH co-varied, with maximum values (O2: 280 µmol l−1, pH: 8.42) in the upper 12–20 m, sharp decreases between 22 and 50 m, slight increases between 50 and 200 m and a drop to minimum values (O2: 91 µmol l−1, pH: 7.71) below 300 m (Fig. 3). A weaker horizontal gradient was evident where sites located at the mouth of the fjord (Lilliguapi and Cross Lilli) displayed higher temperature and pH values than sites in the inner fjord (Figs. 2 and 3). High chlorophyll concentrations were limited to the upper 20 m with a peak at 13 m at the site Near Telele.

Figure 3 Vertical profiles of physico-chemical parameters in Comau Fjord.

Depth profiles at the various ROV stations compiled in Ocean Data View for (A) salinity, (B) temperature, (C) pH and (D) oxygen concentration. The red dotted lines mark the 270–280 m depth range.

Table 2 Physico-chemical environment: site statistics.

Parameter	Site (depth range)	Range	Mean ± SD	
Salinity	LL (1–198)	24.3–32.7	32.3 ± 0.9	
	CL (5–115)	30.8–32.6	32.3 ± 0.4	
	NT (13–422)	31.9–33.0	32.6 ± 0.2	
	SW (1–209)	16.5–32.6	32.1 ± 1.8	
	CH (1–282)	14.4–32.7	32.0 ± 2.5	
	PG (1–355)	NA	NA	
	RB (6–216)	31.0–32.6	32.4 ± 0.2	
Temperature (°C)	LL (1–198)	10.6–15.5	11.2 ± 1.2	
	CL (5–115)	10.6–15.7	11.4 ± 1.3	
	NT (13–422)	10.3–13.4	10.6 ± 0.3	
	SW (1–209)	10.3–16.3	11.0 ± 1.5	
	CH (1–282)	10.2–16.3	10.8 ± 1.4	
	PG (1–355)	10.2–17.0	10.6 ± 1.0	
	RB (6–216)	10.2–15.9	10.5 ± 0.9	
pH	LL (1–198)	7.86–8.33	7.92 ± 0.11	
	CL (5–115)	7.88–8.35	7.95 ± 0.14	
	NT (13–422)	7.74–8.16	7.81 ± 0.04	
	SW (1–209)	7.74–8.34	7.87 ± 0.14	
	CH (1–282)	7.75–8.42	7.87 ± 0.15	
	PG (1–355)	7.72–8.41	7.86 ± 0.11	
	RB (6–216)	7.71–8.37	7.83 ± 0.10	
Oxygen (µmol/l)	LL (1–198)	146–231	158 ± 20	
	CL (5–115)	144–268	160 ± 29	
	NT (13–422)	91–236	140 ± 18	
	SW (1–209)	127–255	161 ± 28	
	CH (1–282)	131–275	160 ± 29	
	PG (1–355)	117–280	148 ± 27	
	RB (6–216)	112–276	149 ± 25	
Notes.

SD standard deviation

NA not available

Substrate

The total surface analysed (Fig. 4A) from the extracted video frames varied between stations and across depths as the deepest depth could not be sampled at all sites due to the fjord bathymetry and the time available to work at each site. The mean surface analysed for each depth also varied as the ROV could not always be kept at a constant distance to the seafloor and some video sequences had to be discarded for technical reasons. In spite of these limitations, the area of non-sediment covered rock suitable for coral settlement (available substrate, Fig. 4B) was not related to the surface analysed (linear regression, r2 < 0.05) so that our sampling can be considered representative of each site and depth.

Figure 4 Substrate distribution in Comau Fjord.

(A) Total surface analysed. (B) Total surface of non-sediment covered rock. (C) Proportion of non-sediment covered rock used/unused by corals. The maximum depth reached during the dives (short grey lines) might be located above the deepest data points which are positioned in the centre of their 10 m depth interval. The red dotted lines mark the 270–280 m depth range.

The available hard substrate (Fig. 4B) was unevenly distributed across the fjord and between depths but non-sediment covered rock suitable for corals was found at all sites and depths considered.

Coral distribution

Desmophyllum dianthus was present at all sites investigated, covering the entire depth range (16–395 m) between the halocline and the fjord bottom. Coral abundances (Fig. 5A) and patch sizes (Fig. 5B) were variable but D. dianthus was found on most of the available hard substrate (Fig. 4C). In spite of the ubiquity of the corals, a vertical pattern emerged, with higher densities of corals in the upper compared to the lower part of the fjord, delimited by the 270–280 m depth lines (two-sample Wilcoxon tests, p < 0.001). Due to the small sample size below 270–280 m depth (N = 2 stations), we performed a permutation test (Efron & Tibshirani, 1993) with 10,000 repetitions which confirmed with a 95% probability that this change in coral abundance is not by chance. The water parameters at 280 m depth are summarized in Table 3.

Figure 5 Coral distribution in Comau Fjord.

(A) Coral abundance. (B) Coral patch size. (C) Coral shape (length/width ratio). The maximum depth reached during the dives (short grey lines) might be located above the deepest data points which are positioned in the centre of their 10 m depth interval. The red dotted lines mark the 270–280 m depth range.

Table 3 Physical oceanography at 280 m depth.

Site	Salinity	Temperature (°C)	pH	Oxygen (µmol/l)	
NT	32.71	10.46	7.83	147.58	
CH	32.71	10.46	7.83	145.98	
PG	NA	10.49	7.84	140.04	
Notes.

NA not available

Coral shape (Fig. 5B) did not show any clear pattern in relation to the environmental gradients. Long, thin corals (L/W > 5) were mostly found in dense coral banks but were also present in less dense areas while short, thick corals (L/W < 3) were present in both poorly and densely populated aggregations. A large patch of mostly elongate corals was found near the river at the head of the fjord (Rio Bodudahue) (Fig. 6), suggesting that parameters other than the ones measured by our ROV may account for the observed shape differences.

Figure 6 Corals in Comau Fjord.

Frames extracted from the ROV videos. (A) Long and thin Desmophyllum dianthus forming a dense bank (Rio Bodudahue, 190 m depth) with some examples of pseudo-colonies. Seen from the side, some corals could be measured when they were not hidden behind other individuals. (B) Dense coral aggregation (Lilliguapi, 124 m depth) with almost all individuals having their tentacles extended. The angle to the substrate was ideal for determining abundances but coral sizes could not be measured. (C) Patch of the bivalve Acesta patagonica Dall, 1902 (Near Telele, 395 m depth), often associated with D. dianthus below 60 m depth. The small group of scleractinian (white) on the lower right corner is the deepest record of D. dianthus during this study. (D) Example of non-colonized sediment-free rock (darker) and sediment-covered substrate (lighter) (Near Telele, 420 m depth). The white scales represent approximately 5 cm in the centre of each frame, where the echo-sounder measured the distance to the substrate.

Regression analyses

The relationships between the coral distribution (coral abundance and substrate use) and the environmental characteristics (temperature, salinity, pH, oxygen and available substrate) were extremely weak (r2 well below 5%) for most parameters (Table 4). When the low-salinity surface layer was excluded from the analyses, then salinity had a weak influence on the coral abundance (r2 about 17%) and the pH, oxygen concentration and substrate availability only slightly explained the percentage of substrate used by the corals (10% < r2 < 17%). Thus, neither pH nor any other parameter measured appears to affect the distribution of D. dianthus in Comau Fjord.

Table 4 Regression analyses.

Coral parameter	Environmental
parameter	r2 (whole
water column)	r2 (excluding
surface layer)	
Abundance	pH	0.010	0.004	
	Oxygen	−0.004	−0.006	
	Temperature	−0.008	0.023	
	Salinity	0.030	0.168	
Substrate usage	pH	−0.006	0.107	
	Oxygen	0.017	0.155	
	Temperature	0.005	−0.006	
	Salinity	0.004	0.059	
	Available substrate	0.160	0.166	

Discussion

Fjord environment

From the physico-chemical structure of Comau Fjord (Figs. 2 and 3) we could identify three water masses. (1) A layer with low salinity, high summer temperatures, pH, oxygen concentration and primary production at the surface (González et al., 2010; Sánchez, González & Iriarte, 2011) results from the large freshwater input into the fjord (Dávila, Figueroa & Müller, 2002). The freshwater outflow is compensated by the inflow of (2) Subantarctic oceanic water (Palma & Silva, 2004; Valle-Levinson et al., 2007) constituting an intermediate layer. Below 300 m we found (3) a water mass with a longer residence time in the fjord (Farmer & Freeland, 1983) showing higher salinity and temperature and lower pH and oxygen due to decomposition processes in the lower water column and fjord bottom (Prado-Fiedler, 2009; Silva, 2008). The 300 m isobath is also the one separating the deep Comau Fjord from the shallower parts of the adjacent Golfo de Ancud (Fig. 1), suggesting a less frequent exchange of these deeper water masses. At the innermost sites, near the head of the fjord, another structure could be detected between the surface and intermediate layers with lower pH and oxygen. This fourth layer could be the consequence of some of the bottom water being washed up by the inflow of oceanic water (Silva, 2008) or the result of a high decomposition after a large phytoplankton bloom.

Controls on coral distribution

Our detailed study confirms anecdotal evidence from the previous shallow ROV survey (Häussermann & Försterra, 2007) that Desmophyllum dianthus may be ubiquitous in Comau Fjord. We found this cold-water coral at virtually all depths below the low salinity surface layer and down to the bottom of the fjord at 395 m (Fig. 6). D. dianthus was also present everywhere between the mouth and the head of the fjord. Most remarkable was the occurrence of a vast coral bank in the direct vicinity of the river at the head of the fjord (Rio Bodudahue), in striking contrast to the alleged sensitivity of this coral to terrigenous sediments (Försterra & Häussermann, 2003).

None of the environmental parameters recorded by our ROV could be identified as a major controller of the scleractinian distribution and, similarly to deep-sea specimens of the same species (Thresher et al., 2011), the shape of D. dianthus individuals in Comau Fjord appeared independent of the depth. Our results suggest that density, pH, oxygen concentration and the quantity of hard substrate available for settlement are less important for D. dianthus than for other CWC species (e.g., Davies et al., 2008; Dullo, Flögel & Rüggeberg, 2008; Fink et al., 2012). Nevertheless, fossil records of D. dianthus in the North Atlantic (Robinson et al., 2007; Thiagarajan et al., 2013), Tasmanian Seamounts (Thiagarajan et al., 2013), and Drake Passage (Burke et al., 2010; Margolin et al., 2013) display changes in the distribution of this scleractinian, most likely due to variations in the surface productivity, the subsurface oxygen concentration and the seawater carbonate chemistry caused by climate-induced modifications of the ocean circulation. The ubiquity of D. dianthus in Comau Fjord demonstrates that no threshold in any of the environmental parameters is crossed there that would completely prevent recruitment, growth or survival of the coral.

The pervasiveness of D. dianthus described above defied identification of an environmental controlling factor. Nevertheless, an important pattern emerged from our quantitative analysis: the significant reduction, in spite of available sediment-free rock, in D. dianthus cover and abundance below 270–280 m, corresponding to the water mass in the deep basin part of the fjord. In the absence of experimental evidence, we can only speculate that the lower oxygen concentration, combined with a lower pH, may entail higher metabolic expenditures for the corals (Dodds et al., 2007; McCulloch et al., 2012). The likely reduced water exchange may in addition limit the supply of food to D. dianthus, which is known to preferably consume the largest size fraction of zooplankton (Mayr et al., 2011), i.e., large copepods and euphausiids (Sánchez, González & Iriarte, 2011). It could also be speculated, in the virtual absence of information on D. dianthus reproduction, larval distribution and behaviour, that the deeper basin parts of Comau Fjord are decoupled from the circulation in the upper part of the water column thus limiting the influx of coral larvae to deep waters.

Desmophyllum, pH and ocean acidification

In Comau Fjord, we found corals within a wide range of pH values (7.71–8.42). Furthermore, Silva (2008) measured a pH of 7.6 there in 1995 and the lowest pH recorded by Jantzen et al. (2013a) in 2011 was 7.4 in the same fjord. Given the difficulties in accurately determining pH (Riebesell et al., 2010) and the current lack of information on residence times in the different water masses, it is difficult to decide to what extent the range of values reported reveal seasonal and/or inter-annual variations in pH, or rather inaccuracies in the different measurements. These uncertainties notwithstanding, it is beyond doubt that Desmophyllum dianthus, is able to cope with large spatial, and likely also temporal, gradients in pH and aragonite saturation state.

In 2011, the aragonite saturation horizon was located at about 150 m depth (Jantzen et al., 2013a). The presence of D. dianthus below this depth appears to be in contradiction with the restriction of cold-water corals to aragonite saturation states >1 (Guinotte et al., 2006). However, Thresher et al. (2011) also found D. dianthus in waters largely undersaturated with regard to aragonite, supporting experimental findings that scleractinian display physiological mechanisms which enable them to increase their internal pH relative to seawater pH, facilitating calcification (Venn et al., 2011). These recent reports, along with the findings of our study, suggest that cold-water corals may be able to cope with ocean acidification better than previously assumed, but likely at a high energy cost (McCulloch et al., 2012). High primary and secondary production in marine ecosystems of northern Patagonia (González et al., 2010), reflected by the high phosphorus content in the skeleton of D. dianthus individuals collected in Chilean fjords (Montagna et al., 2006), could help this coral sustain the cost of calcification under low pH (Cohen & Holcomb, 2009; Jantzen et al., 2013a), at least for the population located above the deep water layer. The individuals growing within the euphotic zone might be exposed to the additional cost of a parasitic relationship with endolithic photo-autotrophs (Hassenrück et al., 2013). It is possible that the corals living in the deepest parts of Comau Fjord might show pre-acclimation or pre-adaptation to life in future lower pH oceans. At present, it cannot be ruled out either that the deeper D. dianthus in the fjord might belong to a genetically divergent population (Miller et al., 2011).

Outlook

Aragonite saturation state is considered as one of the most important drivers of cold-water coral distribution (Davies & Guinotte, 2011). Yet direct measurements of the carbonate chemistry in the vicinity of CWC reefs are very rare (Lunden, Georgian & Cordes, 2013). Furthermore, the few long-term experimental manipulations conducted on CWC showed that at least two of the main reef-building species: Lophelia pertusa and Madrepora oculata remain quite unaffected by pH levels projected for the end of the century (Form & Riebesell, 2012; Maier et al., 2013) and the skeletal density of several CWC species seems independent of the surrounding carbonate saturation (Lunden, Georgian & Cordes, 2013; Thresher et al., 2011). There is thus a great need to acquire more information in order to be able to define the ecological niche of CWC with respect to the seawater carbonate chemistry (Findlay et al., 2013).

Prospering at the lowest limit of seawater acidity and aragonite undersaturation known for cold-water scleractinian in a relatively accessible location, the Desmophyllum dianthus population(s) of Comau Fjord provides a great opportunity to investigate the effect of ocean acidification on CWC.

In order to better encompass all aspects of its inter-annual, seasonal and tidal dynamic, time series of hydrographic and chemical sampling in the fjord are needed. Physiological experiments on the corals would help to determine eventual thresholds for their growth, survival and reproduction, especially when focusing on the deep-dwelling individuals, which might display specific adaptation to low pH. Genetic analyses may even reveal a distinct population, potentially more resistant to aragonite undersaturation.

Many thanks are due to Tobias Funke, our ROV engineer and pilot. We are grateful to Verena Häussermann, Gunter Försterra, Reinhard Fitzek and the Huinay Scientific field station team for their logistic support and to Mauricio Melipillán, our boat captain. We also thank Carin Jantzen and Jürgen Laudien for accompanying this project. Kerstin Jerosch provided support for the georeferencing of the frames. Nils Oswianowski wrote the SciLab routine for the creation of the worldfiles. Milian Noack and Tim Ilic helped with the video analyses. Bathymetry data in Comau Fjord were kindly provided by CN. Patricio Carrasco, Lt. Cdr. Felipe Barrios Burnett, Enrique Silva, Carolina, Calvete and Ortiz Pilar, thank you! We also acknowledge Ruth Alheit for the English review of the manuscript.

This is publication no. 86 of Huinay Scientific Field Station.

Additional Information and Declarations

Competing Interests

Author Contributions

Data Deposition

Laura Fillinger and Claudio Richter declare that they have no competing interests or other interests that might influence the findings reported in this article.

Laura Fillinger and Claudio Richter conceived and designed the experiments, performed the experiments, analyzed the data, contributed reagents/materials/analysis tools, wrote the paper.

The following information was supplied regarding the deposition of related data:

Database: PANGAEA®

Data Publisher for Earth & Environmental Science

http://www.pangaea.de/

Data access: DOI http://dx.doi.org/10.1594/PANGAEA.811911.

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
