# Peer review of "Vertical and horizontal distribution of Desmophyllum dianthus in Comau Fjord, Chile: a cold-water coral thriving at low pH"

_PeerJ, doi:10.7717/peerj.194_

## Round 0.1 · original submission · Minor Revisions

Although reviewers the work is interesting, yet the manuscript at this present form is not publishable yet. They raised a number of issues that need to be carefully addressed. Please prepare your figures in best quality and refer to the figures appropriately in text whenever it is applicable. You may also seek for professional help to tight up the presentation and writing.
When you submit revised version, please provide detailed the responses to the comments on point-to-point format.

Reviewer 1 ·

Basic reporting

I enjoyed reading this interesting and informative study investigating environmental controls on the cold-water coral D. dianthus from the Chilean Fjoird region.

The authors carried out a series of systematic ROV surveys with associated water column analyses to try and put some concrete constraints of where and why these corals can be found. I recommend that this paper be published, with a few changes – particularly to the mesh between text and figures (described below). There does seem to be a lot more that the authors could do with their data in terms of comparative work, but I see that they have tried to make this more a descriptive study. Nevertheless I do point out a few places where they really should look a little more deeply into prior literature.

Throughout the text is would be valuable if attention was drawn to the relevant features of the figures

Since the paper focuses primarily on ASH – then this should be shown on all relevant figures (2, 3, 4, 5). If it was not measured then the best estimate contour of ASH=1 would at least be indicative of when the corals are below the ASH. If this is not known at all then I suggest removing text that says that corals are growing below the ASH – but the text does indicate there is some knowledge of the depth horizon.

I don’t quite see how you picked the 280m depth as the critical depth – is it just site NT? At Site CH it looks shallower? So maybe give a depth range?


The Thresher 2011 CR paper looked at D. dianthus shape – the shape results from that paper could usefully be compared to those results since they are the only two studies (that I know of) that have produced similar data sets.

The study misses the opportunity to compare to other highly relevant papers relating to the controls on D. dianthus. Since the literature on D. dianthus is rather limited it seems that this could readily be addressed – a few suggestions below.

Growth rates of D. dianthus –
Adkins, J. F., et al. (2004). "Growth rates of the deep-sea scleractinia Desmophyllum cristagalli and Enallopsammia rostrata." Earth and Planetary Science Letters 227(3-4): 481-490.

Nutrients and D. dianthus in the Chilean fjords
Montagna, P., et al. (2006). "Phosphorus in Cold-Water Corals as a Proxy for Seawater Nutrient Chemistry." Science 312: 1788-1791.

Environmental controls on D. dianthus
Thiagarajan, N., et al. (2013). "Movement of deep-sea coral populations on climatic timescales." Paleoceanography 28: 1-10 DOI: doi:10.1002/palo.20023.
Fink, H. G., et al. (2012). "Oxygen control on Holocene cold-water coral development in the eastern Mediterranean Sea." Deep-Sea Research Part I-Oceanographic Research Papers 62: 89-96 DOI: 10.1016/j.dsr.2011.12.013.
Burke, A., et al. (2010). "Reconnaissance dating: A new radiocarbon method applied to assessing the temporal distribution of Southern Ocean deep-sea corals." Deep-Sea Research I doi:10.1016/j.dsr.2010.07.010.
Margolin, A., et al. (2013). "Spatial and temporal distribution of scleractinian coral ecosystems from the Drake Passage over the last 35,000 years." Deep-Sea Research. doi.org/10.1016/j.dsr2.2013.06.008.
Robinson, L. F., (2007). "Deep-sea scleractinian coral age and depth distributions in the WN Atlantic for the last 225 thousand years." Bulletin of Marine Sciences 81(3): 371-391.


Figure 2.
Are the ROV dives sites really vertical? It would be really helpful to adapt those ROV sites lines to show coral presence – perhaps breaks in the line where there were no corals?
Watch out for consistency in decimal places.
Label water masses.

Figures 3 and 4.
Include ASH depth.
Include depth of bottom of dive for each location.
What is the difference between a coral bank and a large coral ‘patch’
Out of interest why size and color change for 4b and none of others?

Figure 6
The images I printed out didn’t come out that well – maybe higher resolution versions will be put with the final version?


Small points
I do now know what a worldfile is – more information required.
What do you mean by ‘old’ water?
Line 205 – give depth range
Give both names for ‘Mauri’ in the acknowledgements

Experimental design

Satisfactory, see comments above.

Validity of the findings

Satisfactory, see comments above.

The primary data are available at Pangaea.

·

Basic reporting

This is a very intersting and important scientific contribtion touching a currently severely discussed topic, the potential threat of marine calcifiers to ocean acidification. The authors choose a suitable location where a deep-water coral crosses natural pH gradient. The text is clearly written and enough background information is provided. The cited references cover the story well enough. May I draw the attention to the authors by having a rush on the article cited as:
Thiagarajan N, Gerlach D, Roberts ML, Burke A, McNichol A, Jenkins WJ, Subhas AV, Thresher RE, Adkins JF (2013) Movement of deep-sea coral populations on climatic timescales. Paleoceanography 28:227-236?

Experimental design

No comments of critisim. I recognise that the Comau Fjord is a very remote area to bring marine hi-tech instrumention into operation. Well done.

Validity of the findings

Plausible data collection and statistical treatment with careful interpretation. The authors are well aware of the limitations of their conclusions derived from the methods applied.

Additional comments

Congratulations for your very interesting article. This is the first relevant MS showing the true vertical distribution of D. dianthus within the seep walls of the Comau Fjord. Therefore, this MS is a very useful complementary addition to the Jantzen et al. (2013) paper, already pointing to the pH story.. You tough many potentially limiting factors which may explein presence or absence of D. dianthus except the ageing factor. Have you any idea on growth rates and longevity of your corals studied? I know that this opens another can of worms but you nicely circumvent this topic in your MS. Do we look at mature, young or mixed populations elswhere along your depth transects? Just a thought. Otherwise the MS strongly deserves publication in this journal.

---

## Round 0.2 · accepted · Accept

Dear Dr. Fillinger,
The authors have addressed the issues raised by the reviewers adequately. Therefore, I am ready to accept it.
PY Qian